# Characterization of Lactic Acid Bacteria Isolated from Banana and Its Application in Silage Fermentation of Defective Banana

**DOI:** 10.3390/microorganisms10061185

**Published:** 2022-06-09

**Authors:** Jinsong Yang, Kai Tang, Haisheng Tan, Yimin Cai

**Affiliations:** 1College of Food Science and Engineering, Hainan University, Haikou 570228, China; food868@163.com (J.Y.); a429558911@163.com (K.T.); 2College of Materials Science and Engineering, Hainan University, Haikou 570228, China; ths688@163.com; 3Japan International Research Center for Agricultural Sciences (JIRCAS), Tsukuba 305-8686, Japan

**Keywords:** agricultural by-products, defective bananas, lactic acid bacteria, silage fermentation, tannin

## Abstract

To effectively utilize banana by-products, we prepared silage with defective bananas using screened lactic acid bacteria (LAB), sucrose, and tannase as additives. Eleven strains of LAB were isolated from the fruits and flowers of defective bananas, all of which were Gram-positive and catalase-negative bacteria that produced lactic acid from glucose. Among these LAB, homofermentative strain CG1 was selected as the most suitable silage additive due to its high lactic acid production and good growth in a low pH environment. Based on its physiological and biochemical properties and 16S rRNA gene sequence analysis, strain CG1 was identified as *Lactiplantibacillus plantarum*. Defective bananas contain 74.8–76.3% moisture, 7.2–8.2% crude protein, 5.9–6.5% ether extract, and 25.3–27.8% neutral detergent fibre on a dry matter basis. After 45 d of fermentation, the silage of deficient bananas treated with LAB or sucrose alone improved fermentation quality, with significantly (*p* < 0.05) lower pH and higher lactic acid contents than the control. The combination of LAB and sucrose had a synergistic effect on the fermentation quality of silage. The tannase-treated silage significantly (*p* < 0.05) decreased the tannin content, while the combination of tannase and LAB in silage not only decreased (*p* < 0.05) the tannin content, but also improved the fermentation quality. The study confirmed that defective bananas are rich in nutrients, can prepare good quality silage, and have good potential as a feed source for livestock.

## 1. Introduction

Bananas are mainly produced in Asia, Latin America, and Africa. Global banana production grew at a compound annual growth rate of 3.2% between 2000 and 2017, reaching 114 million tons in 2017 [1]. According to FAO subregional banana production statistics, banana production has continued to increase in recent years, reaching 115 million tons in 2018 and 116 million tons in 2019. More than 135 countries in the world now produce bananas, and some regions rely on bananas as their staple food. As a result, bananas have become one of the world’s most important fruit-food dual-purpose fruits [1]. Bananas are sensitive to weather, and abnormal climate changes in recent years are one of the reasons for the severe reduction in banana production. In a cold winter, when the temperature is below 12 °C, the unripe banana fruit will turn brown affecting its fruit value [2]. During a hot summer, frequent typhoons in tropical regions can hinder the normal growth of bananas, resulting in defects in bananas. In addition, vulnerability to fungal diseases such as fusarium wilt also leads to an increase in the number of defects in banana production [3]. These problems not only affect the yield and quality of bananas, but also have a huge impact on the commercial value of bananas as fresh fruits [4].

Defective bananas are discarded in the field after harvest and are usually disposed of by burning after drying or ploughing into the soil [5,6]. These defective bananas contain a certain amount of moisture and are rich in natural fibre, starch, protein, and mineral components [7,8,9,10]. This makes them suitable for the preparation of silage, which can be used as storage fodder for livestock [11]. Silage fermentation technology may be able to convert waste agriculture by-products into valuable livestock feed, which has important implications for alleviating feed shortages, promoting livestock production, and sustainable agricultural development in the tropics [12].

Silage has the advantages of a strong sour taste and a moist and soft texture, which can improve its palatability to ruminants [13,14,15]. In addition, silage that can be stored can ensure a balanced supply of feed throughout the year and solve the problem of seasonal feed shortages in tropical regions. In recent years, there have been many studies on the use of banana stems and leaves as livestock feed [16,17,18,19], but limited information is currently available on the silage fermentation of defective bananas. Defective bananas are generally small, high in tannins, and low in sugar, which are adverse factors affecting silage fermentation and livestock palatability [20]. Tannase is a tannyl hydrolase that hydrolyses acids with two phenolic groups, such as tannic acid. The enzyme can be produced by moulds such as *Aspergillus niger*, *Aspergillus oryzae*. It can be used to treat tannin and protein in fermented feed to remove astringency, thereby improving the palatability of livestock. Sucrose is a disaccharide with colourless crystals and a sweet taste, which is hydrolysed to produce glucose and fructose. Glucose and sucrose are both important substrates for silage fermentation. Compared with sucrose, some harmful microorganisms related to silage fermentation, which are easier to use glucose for metabolism. Adding sucrose to prepare silage may avoid the vigorous reproduction of some harmful microorganisms and lead to low-quality fermentation. The LAB screened in this study can convert sucrose to lactic acid, therefore sucrose was selected as the sugar addition.

To alleviate feed shortages in tropical regions and promote livestock production, we use banana by-product resources to prepare silage. In order to improve the fermentation quality and livestock palatability of defective banana silage, we also screened LAB from the fruits and flowers of defective bananas as the most suitable silage additive and prepared the defective banana silage with LAB, sucrose, and tannase as additives.

## 2. Materials and Methods

### 2.1. Sample Collection and LAB Isolation

Defective banana fruits and banana flowers were collected at the Wuzhishan Banana Park (Hainan, China) in October 2016. Samples were immediately transported to the laboratory in sterile boxes with ice packs. The samples were cut into approximately 5 mm lengths, and 10 g samples were placed in a vacuum bag containing 90 mL of de Man, Rogosa, and Sharpe (MRS) liquid medium for anaerobic enrichment culturing in an incubator at 30 °C for 24 h. The enriched culture medium was diluted to a concentration of 10^−7^. Three appropriate concentration gradients (10^−4^ to 10^−6^) were selected, and 100 μL aliquots were spread and incubated on the modified MRS agar medium containing 0.5% calcium carbonate under anaerobic conditions for 48 h. Single colonies with distinct calcium carbonate dissolution rings were isolated, cultured, and purified, and the isolated strains were stored in a freezer at −80 °C temperature for future studies. The experimental flow chart is shown in Figure 1.

### 2.2. Physiological and Biochemical Tests of LAB

The test of cell form and physiological and biochemical characteristics of LAB isolates were performed as described by Kozaki et al. [21]. Gram stain was tested involving three processes: staining with a water-soluble dye as crystal violet, decolorizing with ethyl alcohol, and counterstaining with safanin. The catalase reaction test was to add 3% hydrogen peroxide to the cultured cells to observe whether bubbles form.

The test of gas from glucose was to put the liquid medium in the Durham tube, sterilize with high-pressure steam, quench after sterilization, and select a fermentation tube without bubbles in the small glass tube. The LAB strains were inoculated into fermentation tubes and incubated at 35 °C for 48 h. If there are bubbles in the small glass tube after incubation, the inoculated bacteria are judged to have gas-producing abilities. In the gelatine liquefaction test, the test LAB were inoculated into a tube containing 2% gelatine basal medium, cultured at 35 °C for 48 h, and then placed in ice water to observe the inoculation tube. The liquefaction was a positive reaction, and the coagulation was a negative reaction.

The isomers of lactic acid were determined enzymatically with reagents obtained from Biehringer GmbH, Mannheim, Germany. The API (Analytic Products Incorporation) 50 CH method was used to determine the sugar fermentation of LAB. Carbohydrate assimilation and fermentation of 49 compounds and a control were identified on API 50 CH strips (bioMérieux, Tokyo, Japan).

### 2.3. 16S rDNA Sequence Analysis

The 16S rDNA sequence analysis and construction of phylogenetic trees were performed as described previously [22,23]. The 16S rDNA of the strain was amplified by a polymerase chain reaction (PCR) with the bacterial universal primers 27F and 1492R. The PCR program was as follows: 94 °C for 5 min; 35 cycles at 95 °C for 30 s, 55 °C for 30 s, and 72 °C for 1.5 min; and 72 °C for 10 min. The PCR amplification products were purified by 1.0% agarose gel electrophoresis and observed in a gel imager prior to 3730 sequencing. The 16S rRNA gene sequences were compared using GenBank and the BLAST program using the National Center for Biotechnology Information (NCBI) website. The accession numbers of the 16S rDNA sequences used are: Strain CG 1, LC710452; CG2, LC710453; CG3, LC710454; CG4, LC710455; CG5, LC710456; CG6, LC710457; CG7, LC710458; CH1, LC710459; CH2, LC710460; CH3, LC710461; CH4, LC710462.

### 2.4. Silage Experiment Design

Defective banana fruits including peels were harvested at 70–80% maturity and then cut into 10 mm pieces for use as silage materials. The silage treatments were a control, LAB [*Lactiplantibacillus plantarum* CG1, 10^6^ colony forming units (CFU)/g fresh matter (FM)], sucrose (S, 3% of FM), tannase (T, 3 mg/kg of FM), LAB + S and LAB + S + T. Strain CG1 was isolated from defective bananas and screened as a microbial additive for silage preparation because of its high lactic acid production capacity and good growth under low pH conditions. Silages were prepared by a small-scale fermentation system [14,16]. After weighing 30 kg of materials and mixing evenly, approximately 1 kg of silage materials was packed into polyethylene bag silos and the silos were sealed with a vacuum packaging machine. Silages were made in 30 bags per treatment and stored at room temperature (25–33 °C).

The samples were analysed before starting the experiment, and after 3, 5, 7, 10, 20, 30, and 45 d of ensiling [11,16]. Three bags were randomly sampled from each treatment, and the changes in microbial population and pH during fermentation were measured. At 45 d of ensiling, the fermentation quality and chemical composition of silage were also analysed.

### 2.5. Microbial Analysis

Microorganisms including aerobic bacteria, LAB, yeast, and mould were measured using the plate culture method [21]. Using an ultra-clean workbench, a 10 g silage sample was weighed and placed into an Erlenmeyer flask containing 90 mL of 0.85% sterile physiological saline. It was mixed well with a vortex shaker (Vortex Genie2, Scientific Industries, Inc., Bohemia, NY, USA) and diluted ten-fold to 10^−7^. Three different dilutions of 10^−3^, 10^−5^, and 10^−7^ were spread on nutrient agar (NA), potato dextrose agar (PDA), and MRS agar media [13]. The MRS medium was placed in an anaerobic incubator at 30 °C for 48 h; the NA and PDA media were placed in an aerobic incubator at 30 °C for 48–72 h. After the incubation, the agar plates with 30–100 colonies were used for colony counting. The microbial colonies were reported as viable numbers of microorganisms in CFU/g of FM.

### 2.6. Analysis of Silage Fermentation

The silage extracts were extracted with low-temperature water (Cai, 1999) [13]. Silage samples (10 g) were put into a conical flask filled with 90 mL of sterile water; after mixing, they were placed in a refrigerator at 4 °C for 24 h, then the mixture was filtered through sterile gauze, and the supernatant was centrifuged. After the temperature of the supernatant returned to room temperature, the pH values of the samples were measured with a glass electrode pH meter (MP230; Mettler Toledo, Greifensee, Switzerland). The ammonia nitrogen (ammonia-N) content was analysed by using steam distillation of the filtrates (Cai, 1999) [13]. Silage filtrates were centrifuged at 6500× *g* for 5 min at 4 °C, and the supernatants were passed through a 0.45 μm filter. The organic acid content, including lactic acid, acetic acid, propionic acid, and butyric acid were measured by a high-performance liquid chromatography (HPLC) method [Association of Official Agricultural Chemists (AOAC) [24]; column: Agilent ZORBAX SB-Aq, 4.6 mm × 250 mm, 5 μm; mobile phase: 97% 0.02 mol/L KH_2_PO_4_ plus 3% Methanol; flow rate: 1.0 mL/min; column temperature: 30 °C; detection wavelength: 214 nm; injection volume: 10 μL.

### 2.7. Chemical Analysis

Pre-ensiled defective bananas and their silage samples were dried in a forced-air oven (DHG-9203A, Ningbo Jiangnan Instrument Factory, Ningbo, China) at 70 °C for 48 h, then ground to pass through a 1 mm mesh screen (FW 100; Taisite Instrument Co., Ltd., Tianjin, China) for chemical composition analysis. The chemical composition on a dry matter (DM) basis was corrected for residual moisture after 3 h at 105 °C. The DM, ash, crude protein (CP), and ether extract (EE) were analysed by the AOAC methods [24]. The organic matter (OM) content was calculated as the weight loss upon ashing. The neutral detergent fibre (NDF) and acid detergent fibre (ADF) were measured according to the Van Soest method [25] with an ANKOM A200i fibre analyser (ANKOM Technology, Macedon, NY, USA).

The water-soluble carbohydrates (WSC) including glucose, fructose, and sucrose were measured by an HPLC method as described by Cai [26]. The tannin content was determined by phosphomolybdic acid-tungstate sodium colorimetry [20]. The ammonia-N content was determined by phenol-sodium hypochlorite colorimetry [26].

### 2.8. Statistical Analysis

The data were analysed by an analysis of variance (ANOVA) and the means of the samples were then tested for significance using Duncan’s multiple range method. All statistical procedures were performed using the statistical packages for the social sciences (SPSS 13.0 for Windows; SPSS Inc., Chicago, IL, USA). The significance was declined at *p* < 0.05.

## 3. Results

### 3.1. Characteristics and Identification of the LAB Strains Isolated from Defective Bananas

The physiological and biochemical characteristics of the LAB strains isolated from defective bananas are shown in Table 1. A total of 11 strains showed obvious calcium carbonate dissolution rings. Among them, strains CG1, CG2, CG3, CG4, CG5, CG6, and CG7 were isolated from banana fruit, and strains CH1, CH2, CH3, and CH4 were isolated from banana flowers. All strains were Gram-positive and catalase-negative homofermentative bacteria that produced DL-lactic acid from glucose. These strains did not grow at temperatures of 5 °C or 50 °C or at pH 2.5. Strains CG1, CG5, CG7, and CH2 could grow well at temperatures of 10 °C or 45 °C, pH 3.5, and in 3.0% or 6.5% NaCl.

The sugar fermentation and 16S rDNA sequence similarity of LAB isolated from defective bananas are shown in Table 2. Strains CG1, CG5, CG7, and CH2 fermented most of the carbon sources including arabinose, pine triose, and xylose, but not rhamnose. According to the results of 16S rDNA sequence analysis, the 16S rDNA sequence similarities between the isolates and type strain *Lactiplantibacillus plantarum* ATCC14917 were >99.8%, indicating that all isolates were closely related to *L. plantarum*.

### 3.2. Microbial Population and pH Changes of the Defective Bananas during the Ensiling

The microbial population and pH of the defective bananas during the ensiling process are shown in Table 3. After 3 d of fermentation, the LAB counts in the LAB, LAB + S, and LAB + S + T treatment reached 8.2–9.2 log_10_ CFU/g of FM, while the other treatments were lower than 6.4 log_10_ CFU/g of FM. After 5 d of fermentation, the LAB counts showed the highest peak in all treatments and then the LAB counts gradually decreased; the highest numbers of LAB were found in the LAB + S + T-treated silage. After 45 d of fermentation, the counts of LAB in the LAB + S + T, LAB + S, and LAB treatments were higher (*p* < 0.05) than in sucrose, tannase, and control treatments, and the LAB + S + T treatment had the highest LAB count. The number of aerobic bacteria and yeasts increased in each treatment during the first 3 d of ensiling, and then slowly decreased. After 45 d of ensiling, the aerobic bacteria count in LAB + S and LAB + S + T treatments were lower than in other treatments; yeast numbers were in the range of 3.5–4.2 log_10_ CFU/g of FM in all silages. The moulds in all silages were below the detection level of <2.0 log_10_ CFU/g of FM. During the ensiling process, the pH values of all silages continuously decreased.

During the ensiling process, the pH of all silages continuously decreased. The pH in the LAB + S and LAB + S + T treatments decreased rapidly in the early stage of fermentation, and their final pH of terminal fermentation was the lowest, which was 3.70. After 45 d of fermentation, the pH of LAB or sucrose treatment was lower than (*p* < 0.05) LAB + S or LAB + S + T treatment, but higher (*p* < 0.05) than tannase or control treatment.

### 3.3. Fermentation Quality of the Defective Banana Silage

The fermentation quality of the defective banana silage after 45 d of ensiling is shown in Table 4. The moistures of all silages were similar. The lactic acid contents in all additive-treated silages were significantly (*p* < 0.05) higher than in the control silage. Among the silages treated with additives, the LAB + S and LAB + S + T treatments had a higher (*p* < 0.05) lactic acid content than the LAB and sucrose treatments. The acetic acid content of the LAB treatment was lower (*p* < 0.05) than control, tannase, and sucrose treatments, but higher (*p* < 0.05) than in the LAB + S and LAB + S + T treatments. The propionic acid and butyric acid contents of the control, sucrose, and tannase treatments were similar, with those being lower than 0.22% and 0.08% on an FM basis, respectively. The propionic acid and butyric acid contents in LAB, LAB + S, and LAB + S + T treatments were below the detection level (<0.001% of FM). The ammonia-N contents of LAB and sucrose treatments were lower (*p* < 0.05) than in the control while higher (*p* < 0.05) than LAB + S and LAB + S + T treatments.

### 3.4. Chemical Composition of Defective Banana Material and Silage

The chemical composition of defective banana material before ensiling is shown in Table 5. Before ensiling, the moistures in all treatments were 74.8–76.3%. Their OM, CP, EE, NDF, and ADF contents were 93.2–93.8%, 7.2–8.2%, 5.9–6.5%, 25.3–27.8%, and 17.5–18.9% on a DM basis, respectively. The WSC content of defective bananas was 16.3–18.9% of DM in sucrose, LAB + S, and LAB + S + T treatments, which was higher (*p* < 0.05) than in the control and LAB treatment. The tannin contents in materials were approximately 1.6–1.8% of DM, with no significant differences among treatments.

The chemical composition of defective banana silage after 45 d of ensiling is shown in Table 6. After 45 d of ensiling, the OM contents of all silages were approximately 93% of DM. The CP and EE contents ranged from 6.3% to 9.3% and 5.7% to 6.4% of DM, respectively. The CP and WSC contents in the LAB + S and LAB + S + T-treated silages were higher (*p* < 0.05) than in other silages. The NDF and ADF contents of all silages were similar. Silage treated with tannase and LAB + S + T had lower (*p* < 0.05) tannin contents than the other silages.

## 4. Discussion

Bananas are the most traded and consumed fruit in the world. In some tropical countries, bananas have become one of the pillar industries of agriculture and play an important role in the economic and rural social development of tropical regions. After the banana fruit is harvested, large amounts of by-products are produced, including banana stems and leaves. In recent years, due to the influence of global warming, a large number of defective bananas are produced every year. Usually, these by-products are discarded in the fields as waste, which not only wastes resources but also pollutes the environment, and spreads latent germs, negatively impacting on production [4,5]. Studies have shown that banana stems, leaves, skins, and other wastes are rich in protein, sugar, vitamins and other nutrients, and can be used as raw materials for livestock feed. However, because these by-products are prone to spoilage due to their high moisture content, silage is considered an important fermentation technology for the efficiently utilization of these resources [27].

The LAB play an important role in the silage fermentation process. The LAB and other environmental microorganisms are often found living in association with plant material including forage crops, grasses, vegetables, fruits, and silages [14,16]. Microorganisms harmful to silage fermentation, including aerobic bacteria and moulds, are often sensitive to anaerobic and acidic environments [28,29]. In particular, some Gram-negative aerobic bacteria such as coliform bacteria have the characteristics of thin cell walls and weak acid resistance and will die rapidly under the influence of anaerobic and acidic environments formed by silage fermentation. However, LAB can proliferate vigorously in anaerobic and acidic silage environments, quickly succeed as dominant bacteria, and promote lactic acid fermentation. As shown in Table 3, in the first 3–5 days of ensiling, the LAB became the dominant community in the LAB alone or mixed treatments and dominated the silage fermentation. In addition, as can be seen from Table 4, the lactic acid content increased and the ammonia nitrogen content decreased in these LAB-treated silages, a fact that was also verified from these final fermentation products.

Epiphytic LAB of fresh forages are rich in diversity. The LAB related to silage fermentation can be classified into lactic acid-producing rods such as *Lactobacillus* species and lactic acid producing cocci such as *Weissella*, *Lactococcus*, *Leuconostoc*, *Enterococcus*, and *Pediococcus* species in terms of cell morphology [11]. Generally, coccid LAB is weak in acid resistance and likes to grow in a slightly aerobic environment. Some cocci are heterofermentative types and grow vigorously in the early stage of silage fermentation, producing carbon dioxide and creating an anaerobic environment for the subsequent fermentation of other acid-tolerant lactobacilli. Our previous studies have reported that LAB is the main microbial community in fruit residues, and previous isolates from fruits have been identified as *L. plantarum*, *Lactobacillus casei*, and *Pediococcus pentosaceus* [13,14,16]. In the present study, the isolates from defective bananas were Gram-positive and catalase-negative rods that did not produce gas from glucose and formed DL-lactic acid. These properties show that these strains belong to the genus *Lactiplantibacillus*. The 16S rDNA sequence similarity between all isolates and the type strain *L. plantarum* ATCC14917 was >99.8%, which confirmed that these strains belonged to the genus *Lactiplantibacillus* and were most closely related to *L. plantarum*. In addition, these strains were similar to the type strain *L. plantarum* ATCC14917 in some carbohydrate fermentation patterns, such as those of glucose, fructose, lactose, and galactose. Two isomers of lactic acid, L-lactic acid, and D-lactic acid are produced by LAB in silage fermentation. In animals, L-lactic acid is more effectively utilized within the body than D-lactic acid, while D-lactic acid has no direct neurotoxic effect on livestock [11,30]. In the production of livestock feed, L-lactic acid producing Lactobacillus casei and DL-lactic acid-producing Lactiplantibacillus plantarum are widely used to develop the commercial inoculant [11,14]. The LAB screened in this study is Lactiplantibacillus plantarum, which forms DL-lactic acid. In the process of silage fermentation, this LAB can produce a large amount of lactic acid and reduce pH during ensiling, thereby improving the fermentation quality of silage.

Generally, silage is based on natural lactic acid fermentation [13]. Moisture, LAB, and WSC are important factors affecting silage fermentation. In addition, aerobic bacteria, yeast and mould all coexist in the silage environment, which are harmful microorganisms for silage fermentation. The moisture content of defective bananas in this experiment was as high as 75%, which exceeded the ideal range of 60–70% moisture content for silage preparation. In addition, the number of epiphytic LAB showed a level of 103 CFU/g of FM, which was lower than the basic bacteria number 105 required for high-quality silage preparation [31,32]. When LAB fail to produce sufficient lactic acid during fermentation to reduce the pH and inhibit the growth of harmful bacteria, the resulting silage will be of poor quality. As shown in Table 3, low numbers of LAB and high numbers of aerobic bacteria and yeasts were present in the banana material, resulting in a poor-fermentation quality in the control silage. The factors involved in assessing the fermentation quality of silage include the microbial structure and the WSC of the silage material. Defective bananas have relatively low LAB counts and WSC content. During ensiling, the LAB could not produce sufficient lactic acid to improve silage fermentation. Therefore, it is necessary to screen and use microbial inoculants to control the silage fermentation of defective bananas.

In the present study, the defective banana silages treated with LAB, sucrose, and their combination were well preserved, with high lactic acid contents and low pH values. This may be related to the properties of the screened LAB, which can grow well under low pH conditions. During ensiling, these LAB can produce enough lactic acid to lower the pH and inhibit the growth of harmful bacteria; therefore, the resulting silage was of good quality. In addition, the defective banana raw materials used in this study are natural fruit by-products without any added antibacterial substances or preservatives. They are completely prepared through the principle of silage fermentation and ingeniously utilize the fermentation function of LAB. Our follow-up experiments confirmed that the shelf life of defective banana silage can be more than 1 year under the premise of well-maintained anaerobic conditions. It is well known that the shortage of feed in the dry season is a major factor limiting the development of animal husbandry in the tropics. In tropical countries and regions, the main sources of ruminant feed are native grasses and crop by-products [32,33,34]. With the increase of population, the decrease of arable land and the development of animal husbandry, the production of forage crops and grasses cannot meet the needs of livestock raising [33,34,35]. The development of new natural feed resources such as banana stems and defective bananas, and their utilization in livestock production have become an important strategy to cope with the worldwide feed shortage.

The main sources of feed for ruminants such as cattle, sheep, and goats are grasses, forage crops, crop straws, grains, and their by-products. Due to the rising prices of these raw materials, the cost of feed is increasing. Therefore, cheap fruit by-products are gradually being used as raw materials for producing feed [16]. Banana by-products are an abundant resource that is rich in nutrients and has obvious advantages as an animal feed resource [14]. In this study, the defective bananas contained more than 7% CP, 5% ether extract, 25% NDF, and 16% WSC, indicating that the defective bananas can be used as a source of livestock feed. In addition to CP and fat, they are also rich in plant dietary fibre, minerals, and vitamins. The content of these useful components in defective bananas, the dynamic changes during silage fermentation, and the response to the physiological metabolism of livestock should be further explored in future studies. According to our previous analysis of tea, coffee, vegetable, and fruit residue silage, they can be used as a high-quality raw material source for the total mixed ration preparation, and the dry matter mixing ratio can generally be adjusted below 20% [36,37,38,39]. Therefore, the defective banana silage can replace part of the roughage or concentrate diet according to the nutritional needs of feeding livestock, which can play a certain role in effectively utilising fruit by-product resources and reducing feed costs. However, the tannin (tannic acid) component of banana by-products not only affects the palatability to animals, but also inhibits the activity of rumen microbial enzymes and affects the digestibility. In the present study, both the defective banana material and the control silage had a higher tannin content. When tannase was added to the silage, the tannin content was significantly reduced. The combined addition of tannase and LAB to silage had a synergistic effect of reducing tannin and improving the fermentation quality. This is because tannins are degraded by tannase during fermentation, which not only inhibits the growth of harmful bacteria, but also promotes silage fermentation. This is of great significance to improve the palatability, digestibility of rumen protein, and feeding value of defective banana silage. In addition, the LAB community in silage also plays an important role in the physiological metabolism and health maintenance of livestock [40,41].

The results of our study demonstrate that high-quality defective banana silage can be prepared for animal production through the bacterial-enzyme synergy of LAB and tannase. This is of great significance for effectively utilizing pomace resources, alleviating feed shortages, and promoting animal husbandry.

## 5. Conclusions

To effectively utilise banana by-products, we isolated LAB from banana fruits and flowers, and screened out the homofermentative Lactobacillus plantarum CG1 strain with high lactic acid production and good growth in a low pH silage environment. The defective banana silages were prepared using selected LAB, sucrose, and tannase as additives. The combination of LAB and tannase showed the synergistic effect, which not only reduced the tannin content and astringency but also improved the fermentation quality of defective banana silage. The results confirm that defective bananas are nutritious, can be used to prepare high-quality silage, and have good potential as a feed source for ruminants. This is of great significance for effectively utilising biological resources, reducing feed costs, alleviating feed shortages, and promoting the sustainable development of animal husbandry.

## Figures and Tables

**Figure 1 microorganisms-10-01185-f001:**
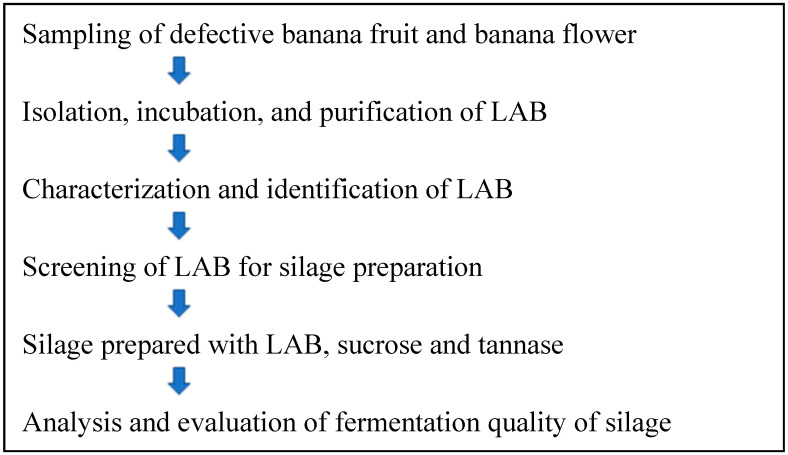
Experimental flow chart of this study.

**Table 1 microorganisms-10-01185-t001:** Physiological and biochemical characteristics of lactic acid bacteria (LAB) strain isolated from defective bananas.

Item	LP	LAB Strain from Defective Banana Fruit							LAB Strain from Defective Banana Flower			
		CG1	CG2	CG3	CG4	CG5	CG6	CG7	CH1	CH2	CH3	CH4
Cell form	Rod	Rod	Rod	Rod	Rod	Rod	Rod	Rod	Rod	Rod	Rod	Rod
Gram stain	+	+	+	+	+	+	+	+	+	+	+	+
Catalase reaction	-	-	-	-	-	-	-	-	-	-	-	-
Gelatin liquefaction	-	-	+	+	+	-	-	-	+	-	+	+
Gas from glucose	-	-	-	-	-	-	-	-	-	-	-	-
Fermentation type	HM	HM	HM	HM	HM	HM	HM	HM	HM	HM	HM	HM
Isomer of lactic acid	DL	DL	DL	DL	DL	DL	DL	DL	DL	DL	DL	DL
Lactate production (%)	1.47	1.60	1.18	1.45	1.28	1.50	1.30	1.52	1.22	1.53	1.35	1.25
Final pH in MRS broth	3.60	3.40	4.50	3.65	4.25	3.50	4.10	3.50	4.30	3.50	3.85	4.20
Growth at temperature												
5 °C	-	-	-	-	-	-	-	-	-	-	-	-
10 °C	+	+	w	+	+	+	w	+	+	+	+	w
45 °C	+	+	+	-	w	+	w	+	w	+	w	-
50 °C	-	-	-	-	-	-	-	-	-	-	-	-
Growth at NaCl												
3.0%	+	+	+	w	+	+	+	+	+	+	+	+
6.5%	+	+	+	-	w	+	w	+	-	+	-	+
Growth at pH												
2.5	-	-	-	-	-	-	-	-	-	-	-	-
3	-	-	-	-	-	-	-	-	-	w	-	-
3.5	+	+	-	w	w	+	-	+	w	+	-	w
4	+	+	-	+	w	+	w	+	w	+	+	w
7	+	+	+	+	+	+	+	+	+	+	+	+

LP, type strain *Lactiplantibacillus plantarum* ATCC14917; +, positive reaction; -, negative reaction; w, weakly positive reaction; HM, homo-fermentation type; DL, DL-lactic acid.; MRS, de Man, Rogosa, and Sharpe medium.

**Table 2 microorganisms-10-01185-t002:** Sugar fermentation of lactic acid bacteria and the similarity of 16S rDNA sequence between the isolate and type strain.

Item	LP	Strain from Fruit of Defective Banana							Strain from Flower of Defective Banana			
		CG1	CG2	CG3	CG4	CG5	CG6	CG7	CH1	CH2	CH3	CH4
Sugar fermentation												
Glucose	+	+	+	+	+	+	+	+	+	+	+	+
Fructose	+	+	+	+	+	+	+	+	+	+	+	+
Lactose	+	+	+	+	+	+	+	+	+	+	+	+
Galactose	+	+	+	+	+	+	+	+	+	+	+	+
Gluconate	+	+	+	+	+	+	+	+	+	+	+	+
Maltose	+	+	+	+	+	+	+	+	+	+	+	+
L-Arabinose	+	w	w	w	w	w	+	+	+	+	+	+
L-rhamnose monohydrate	-	-	-	-	-	-	-	-	-	-	-	-
Melezitose	w	+	+	+	w	w	w	w	w	w	w	w
Melibiose	+	+	+	+	+	+	+	+	+	+	+	+
Aldohexose	+	+	+	+	w	+	+	+	+	+	+	+
Ribose	+	+	w	+	+	+	+	+	+	+	+	+
D(-)-Salicin	+	+	+	+	+	+	+	+	+	+	+	+
Xylose	-	w	w	w	w	w	w	w	w	w	w	w
D(+)-Sucrose	+	+	+	+	+	+	+	+	+	+	+	+
Trehalose	+	+	+	+	+	+	+	+	w	+	+	+
Sorbitol	+	+	+	+	+	+	w	+	+	+	+	+
Mannitol	+	+	+	+	+	+	+	+	+	+	+	w
Raffinose	+	+	+	+	+	+	+	+	+	+	w	+
Laetrile	+	+	+	w	+	+	+	+	+	+	+	+
Cellobiose	+	+	+	+	w	w	+	+	+	+	+	+
Heptaside	+	+	+	+	+	+	+	+	+	+	+	+
Similarity of 16S rDNA sequence (%)	100	99.8	99.8	99.9	99.8	99.8	99.8	100	99.8	99.8	99.9	99.8

LP, type strain *Lactiplantibacillus plantarum* ATCC14917; +, positive reaction; -, negative reaction; w, weakly positive reaction. Similarity of 16S rDNA sequence is that between the isolates with the type strain *Lactiplantibacillus plantarum* ATCC14917.

**Table 3 microorganisms-10-01185-t003:** Microbial population and pH value of the defective bananas during ensiling process.

Ensiling (d)	Silage Treatment					
	Control	LAB	Sucrose (S)	Tannase (T)	LAB + S	LAB + S + T
	Lactic acid bacteria (log_10_ CFU/g of FM)					
0	3.08 ± 0.45 ^b^	6.67 ± 0.76 ^a^	3.15 ± 0.53 ^b^	3.28 ± 0.47 ^b^	6.30 ± 0.55 ^a^	6.65 ± 0.67 ^a^
3	5.43 ± 0.45 ^c^	8.21 ± 0.72 ^b^	6.36 ± 0.63 ^bc^	5.41 ± 0.54 ^c^	8.94 ± 0.68 ^a^	9.16 ± 0.56 ^a^
5	6.53 ± 0.54 ^b^	8.57 ± 0.57 ^a^	7.36 ± 0.47 ^ab^	6.54 ± 0.65 ^b^	8.72 ± 0.71 ^a^	8.75 ± 0.57 ^a^
7	5.78 ± 0.57 ^c^	7.34 ± 0.45 ^ab^	6.71 ± 0.43 ^b^	5.76 ± 0.67 ^c^	8.51 ± 0.52 ^a^	8.48 ± 0.48 ^a^
10	5.58 ± 0.55 ^c^	6.66 ± 0.65 ^b^	6.51 ± 0.51 ^b^	5.25 ± 0.48 ^c^	7.41 ± 0.42 ^a^	7.32 ± 0.45 ^a^
20	4.38 ± 0.43 ^c^	6.57 ± 0.56 ^a^	5.82 ± 0.50 ^b^	4.15 ± 0.53 c	6.18 ± 0.81 ^a^	6.25 ± 0.62 ^a^
30	4.18 ± 0.42 ^b^	5.72 ± 0.57 ^a^	5.34 ± 0.54 ^a^	4.33 ± 0.42 ^b^	5.74 ± 0.73 ^a^	5.85 ± 0.85 ^a^
45	3.81 ± 0.42 ^b^	5.45 ± 0.68 ^a^	5.22 ± 0.52 ^a^	3.72 ± 0.48 ^b^	5.36 ± 0.53 ^a^	5.54 ± 0.56 ^a^
	Aerobic bacteria (log_10_ CFU/g of FM)					
0	6.40 ± 1.22	6.15 ± 0.72	6.33 ± 1.53	5.95 ± 0.56	6.16 ± 0.84	6.72 ± 0.68
3	7.12 ± 0.61	6.82 ± 1.32	7.25 ± 0.75	7.00 ± 1.65	6.76 ± 1.15	6.98 ± 0.87
5	6.41 ± 0.46 ^a^	5.84 ± 0.58 ^b^	6.32 ± 0.63 ^a^	6.43 ± 0.48 ^a^	5.32 ± 0.42 ^b^	5.35 ± 0.53 ^b^
7	6.20 ± 0.56 ^a^	5.05 ± 1.46 ^b^	5.95 ± 0.68 ^a^	5.72 ± 0.75 ^a^	5.12 ± 0.52 ^b^	5.10 ± 0.46 ^b^
10	5.45 ± 0.44 ^a^	4.96 ± 0.53 ^ab^	5.54 ± 0.42 ^a^	5.27 ± 0.74 ^a^	4.83 ± 0.83 ^b^	4.45 ± 0.58 ^b^
20	5.32 ± 0.56 ^a^	4.50 ± 0.74 ^b^	4.36 ± 0.63 ^b^	5.34 ± 0.43 ^a^	4.23 ± 0.47 ^b^	4.00 ± 0.71 ^b^
30	4.71 ± 0.74 ^a^	4.46 ± 0.65 ^a^	4.18 ± 0.48 ^a^	4.73 ± 0.43 ^a^	3.68 ± 0.62 ^b^	3.72 ± 0.54 ^b^
45	3.49 ± 0.48 ^a^	3.51 ± 0.51 ^a^	3.35 ± 0.54 ^a^	3.52 ± 0.62 ^a^	2.64 ± 0.68 ^b^	2.82 ± 0.81 ^b^
	Yeast (log_10_ CFU/g of FM)					
0	5.70 ± 1.46	6.16 ± 0.54	6.15 ± 1.16	6.50 ± 0.65	5.86 ± 1.24	6.25 ± 0.86
3	7.56 ± 1.32	7.30 ± 084	7.89 ± 0.67	7.15 ± 1.53	7.38 ± 0.71	7.86 ± 1.25
5	6.86 ± 0.53	6.42 ± 1.20	7.10 ± 1.38	6.35 ± 0.67	6.70 ± 0.75	7.16 ± 1.54
7	6.76 ± 0.58	5.89 ± 0.65	6.68 ± 0.71	6.78 ± 0.68	5.62 ± 0.48	6.15 ± 0.55
10	5.94 ± 1.12	6.13 ± 0.83	5.45 ± 1.25	6.24 ± 0.56	5.32 ± 0.67	6.10 ± 0.46
20	5.23 ± 0.56	4.51 ± 0.46	4.82 ± 0.43	5.36 ± 0.65	4.91 ± 0.57	4.04 ± 0.44
30	5.11 ± 0.45	4.58 ± 0.64	4.82 ± 0.66	5.14 ± 0.53	4.91 ± 0.59	4.74 ± 0.83
45	4.21 ± 0.57	3.85 ± 0.73	3.52 ± 0.48	4.10 ± 0.45	3.70 ± 1.13	3.56 ± 0.65
	pH					
0	6.18 ± 0.68	6.40 ± 1.12	6.34 ± 0.84	6.58 ± 1.43	6.25 ± 0.62	6.10 ± 0.76
3	5.75 ± 0.75 ^a^	4.80 ± 0.47 ^b^	5.18 ± 0.54 ^ab^	5.53 ± 0.65 ^a^	3.94 ± 0.49 ^c^	4.04 ± 0.45 ^c^
5	5.61 ± 0.61 ^a^	4.72 ± 0.72 ^b^	5.09 ± 0.51 ^ab^	5.50 ± 0.46 ^a^	3.88 ± 0.58 ^c^	3.95 ± 0.53 ^c^
7	5.54 ± 0.53 ^a^	4.60 ± 0.61 ^b^	4.83 ± 0.73 ^b^	5.44 ± 0.52 ^a^	3.83 ± 0.46 ^c^	3.86 ± 0.43 ^c^
10	5.52 ± 0.57 ^a^	4.50 ± 0.51 ^b^	4.76 ± 0.67 ^b^	5.41 ± 0.42 ^a^	3.75 ± 0.73 ^c^	3.80 ± 0.45 ^c^
20	5.37 ± 0.54 ^a^	4.48 ± 0.42 ^b^	4.62 ± 0.53 ^b^	5.40 ± 0.46 ^a^	3.66 ± 0.63 ^c^	3.78 ± 0.54 ^c^
30	5.26 ± 0.56 ^a^	4.32 ± 0.45 ^b^	4.48 ± 0.48 ^b^	5.26 ± 0.52 ^a^	3.75 ± 0.42 ^c^	3.72 ± 0.51 ^c^
45	5.00 ± 0.43 ^a^	4.25 ± 0.52 ^b^	4.43 ± 0.46 ^b^	5.06 ± 0.61 ^a^	3.70 ± 0.55 ^c^	3.70 ± 0.56 ^c^

^a–c^ Data are means of three samples, means in the same column followed by different letters differ (*p* < 0.05). LAB, lactic acid bacteria; Mould did not detect in all silages during ensiling.

**Table 4 microorganisms-10-01185-t004:** Fermentation quality of the defective banana silage after 45 d of ensiling.

Items	Silage Treatment					
	Control	LAB	Sucrose (S)	Tannase (T)	LAB + S	LAB + S + T
Moisture (%)	75.38 ± 1.35	75.24 ± 1.10	74.82 ± 1.64	75.42 ± 1.17	76.12 ± 0.84	75.06 ± 1.22
Lactic acid (% of FM)	0.24 ± 0.12 ^c^	1.35 ± 0.53 ^b^	1.26 ± 0.62 ^b^	0.23 ± 0.13 ^c^	2.20 ± 0.74 ^a^	2.03 ± 0.68 ^a^
Acetic acid (% of FM)	1.06 ± 0.61 ^a^	0.78 ± 0.34 ^b^	0.94 ± 0.45 ^a^	1.12 ± 0.53 ^a^	0.35 ± 0.12 ^c^	0.28 ± 0.10 ^c^
Propionic acid (% of FM)	0.22 ± 0.08	ND	0.19 ± 0.14	0.17 ± 0.06	ND	ND
Butyric acid (% of FM)	0.04 ± 0.01	ND	0.05 ± 0.01	0.08 ± 0.02	ND	ND
Ammonia-N (g/kg of FM)	1.35 ± 0.64 ^a^	0.69 ± 0.38 ^b^	0.81 ± 0.53 ^b^	1.42 ± 0.74 ^a^	0.35 ± 0.10 ^c^	0.39 ± 0.12 ^c^

^a–c^ Data are means of three samples, means in the same raw followed by different letters differ (*p* < 0.05). LAB, lactic acid bacteria; ND, not detected.

**Table 5 microorganisms-10-01185-t005:** Chemical composition of defective banana material before ensiling.

Items	Defective Banana Treatment					
	Control	LAB	Sucrose (S)	Tannase (T)	LAB + S	LAB + S + T
Moisture (%)	75.20 ± 1.32	74.81 ± 1.52	75.52 ± 1.06	76.24 ± 0.85	75.19 ± 1.12	75.36 ± 0.78
Organic matter (% of DM)	94.12 ± 0.94	93.85 ± 1.32	93.64 ± 1.53	94.05 ± 1.28	93.25 ± 1.47	93.80 ± 1.15
Crude protein (% of DM)	7.23 ± 1.15	7.64 ± 0.67	7.34 ± 1.54	7.85 ± 0.93	7.35 ± 1.26	8.11 ± 0.85
Ether extract (% of DM)	6.12 ± 1.62	6.25 ± 0.76	5.98 ± 1.25	6.48 ± 0.83	6.34 ± 0.64	6.45 ± 0.57
NDF (% of DM)	27.76 ± 1.54	26.64 ± 1.23	27.55 ± 1.48	25.38 ± 0.84	27.22 ± 1.82	26.87 ± 1.13
ADF (% of DM)	18.62 ± 1.22	18.94 ± 0.96	18.36 ± 0.85	17.56 ± 1.52	18.35 ± 0.73	17.81 ± 0.78
WSC (% of DM)	16.75 ± 0.76 ^b^	16.34 ± 1.62 ^b^	18.34 ± 0.73 ^a^	16.84 ± 0.86 ^b^	18.43 ± 1.43 ^a^	18.86 ± 0.82 ^a^
Tannin (% of DM)	1.73 ± 0.35	1.58 ± 0.40	1.65 ± 0.44	1.69 ± 0.67	1.75 ± 0.54	1.62 ± 0.32

^a,b^ Data are means of three samples, means in the same raw followed by different letters differ (*p* < 0.05). LAB, lactic acid bacteria; DM, dry matter; WSC, water soluble carbohydrates; NDF, neutral detergent then the analysis was done; ADF, acid detergent fibre.

**Table 6 microorganisms-10-01185-t006:** Chemical composition of defective banana silage after 45 d of ensiling.

Items	Silage Treatment					
	Control	LAB	Sucrose (S)	Tannase (T)	LAB + S	LAB + S + T
Organic matter (% of DM)	93.16 ± 1.42	93.80 ± 2.24	93.36 ± 1.07	93.50 ± 1.13	93.24 ± 0.98	93.72 ± 1.84
Crude protein (% of DM)	6.33 ± 0.56 ^c^	7.08 ± 0.83 ^b^	6.94 ± 0.72 ^b^	6.55 ± 0.58 ^c^	8.92 ± 0.65 ^a^	9.26 ± 0.68 ^a^
Ether extracts (% of DM)	5.72 ± 0.48 ^b^	5.85 ± 0.56 ^ab^	6.06 ± 0.52 ^ab^	6.13 ± 0.61 ^ab^	6.27 ± 0.45 ^a^	6.38 ± 0.72 ^a^
NDF (% of DM)	25.76 ± 1.62	25.63 ± 1.26	26.27 ± 1.08	24.35 ± 1.02	25.57 ± 0.78	24.82 ± 0.84
ADF (% of DM)	18.55 ± 1.47	18.83 ± 1.11	18.32 ± 1.22	17.52 ± 1.04	18.24 ± 1.76	17.73 ± 0.88
WSC (% of DM)	15.63 ± 1.53 ^d^	15.37 ± 1.03 ^c^	16.12 ± 1.06 ^b^	15.85 ± 0.85 ^c^	16.26 ± 1.37 ^a^	16.58 ± 1.65 ^a^
Tannin (% of DM)	1.38 ± 0.55 ^a^	1.20 ± 0.45 ^a^	1.18 ± 0.63 ^a^	0.78 ± 0.38 ^b^	0.86 ± 0.35 ^b^	0.62 ± 0.30 ^b^

^a–d^ Data are means of three samples, means in the same raw followed by different letters differ (*p* < 0.05). LAB, lactic acid bacteria; DM, dry matter; WSC, water soluble carbohydrates; NDF, neutral detergent fibre; ADF, acid detergent fibre.

## Data Availability

Not applicable.

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
