# Peer review of "Characterization of Lactic Acid Bacteria Isolated from Banana and Its Application in Silage Fermentation of Defective Banana"

_microorganisms, 2022, doi:10.3390/microorganisms10061185_

Round 1
Reviewer 1 Report
The work is inserting, valuable, well designed and well written.
Major concern
The most dangerous point in this work is the final approval for strains to be used in silage as D-Lactate has a direct neurotoxic effect and various health problems please kindly refer to https://www.hindawi.com/journals/bmri/2020/3419034/
The authors should deeply discuss this point and confirm the validity of the final end product, otherwise, this Silage product would be toxic.
Minor comment
Please update the data at line 36-37
Author Response
Dear Editor and Reviewers
Thank you very much for evaluating our paper.
We have revised the manuscript “Characterization of Lactic Acid Bacteria Isolated from Banana and Application for the Silage Fermentation of Defective Banana” carefully based on the comments of Editor and Reviewers.
In this revised version, changes to our manuscript were all highlighted within the document by using red colored text. We hope that our paper much better quality than before.
The corresponding responses to the reviewers are as followings:
Reviewer 1
The article “Characterization of Lactic Acid Bacteria Isolated from Banana and Application for the Silage Fermentation of Defective Banana” is focused on studying the efficiency of lactic fermentation of banana by-products (defective banana) in order to obtain silage.
The topic of the paper is of relevance due to the current increasing trend of valorization of different by-products from fruit production and processing.
The study presents numerous analyses and results but the submitted manuscript revealed some inconsistencies which have to be corrected before publication.
Response: Thank you for evaluating our paper. We will do our best to revise this manuscript.
- The aim of the manuscript needs to be rephrased to better indicate the research purpose.
Response: Yes, we have rewritten the purpose according to your comments in the introduction section.
- It is not clearly explained why the authors used sucrose and tannase in their experiment. I suggest introducing an explanatory paragraph to the Introduction section. Moreover, Lb.plantarum did not metabolize very well sucrose, especially when the substrate contain other simple sugars.
Response: Yes, based on your suggestion, we clearly explained why sucrose and tannase were used in the experiment in the introduction section.
- I suggest introducing to the Materials and Methods section a graphical scheme of the experiment to be easier understood by readers.
Response: Yes, we have added the experimental process to the experimental section, hoping to better understand the experimental method.
- Some results are poorly discussed; for example fiber content
Response: Yes, bananas are rich in nutrients. In addition to CP and fat, they are also rich in plant dietary fiber, minerals and vitamins. The content of these useful components in bananas, the dynamic changes during silage fermentation and the response to the physiological metabolism of livestock are necessary to be further explored in future studies. We have addressed this in the discussion section.
- As is shown in Table 3, during ensiling not only LAB but also yeasts and aerobic bacteria are present. Are these microorganisms involved in fermentation? For example, yeast count is relatively high during the first 10 days; yeast cells
Response: Thank you for your very important question. Yes, these microorganisms are involved in the fermentation process and are harmful microorganisms for silage fermentation. We describe it in more detail in the Discussion section.
Minor observations:
- lines 77-78-please correct the word temperature
Response: Yes, we have revised.
- line 82 please correct the word character
Response: Yes, we have revised.
- line 96 is Boehringer the correct word?
Response: Yes, we have confirmed that it is correct.
- line 70-please indicate the used ratio between defective banana fruits and banana flowers
Response: Defective bananas are immature and the fruit and flower parts have not been separated. The proportion of flowers and fruits cannot be specified accurately.
- line 130-please indicate producer, city, country for vortex shaker
Response: Yes, we have added the information in the test.
- line 153-please indicate producer, city, country for the oven
Yes, we have added the information in the text.
- line 270-please correct the word esiling
Response: Yes, we have revised as ensiling.
- line 346-please correct the word improve
Response: Yes, we have revised as improve.

Reviewer 2 Report
The article “Characterization of Lactic Acid Bacteria Isolated from Banana and Application for the Silage Fermentation of Defective Banana” is focused on studying the efficiency of lactic fermentation of banana by-products (defective banana) in order to obtain silage.
The topic of the paper is of relevance due to the current increasing trend of valorization of different by-products from fruit production and processing.
The study presents numerous analyses and results but the submitted manuscript revealed some inconsistencies which have to be corrected before publication.
Some major observations are:
1. The aim of the manuscript needs to be rephrased to better indicate the research purpose
2. It is not clearly explained why the authors used sucrose and tannase in their experiment. I suggest introducing an explanatory paragraph to the Introduction section. Moreover, Lb.plantarum did not metabolize very well sucrose, especially when the substrate contain other simple sugars.
3. I suggest introducing to the Materials and Methods section a graphical scheme of the experiment to be easier understood by readers.
4. Some results are poorly discussed; for example fiber content
5. As is shown in Table 3, during ensiling not only LAB but also yeasts and aerobic bacteria are present. Are these microorganisms involved in fermentation? For example, yeast count is relatively high during the first 10 days; yeast cells are not consuming carbohydrates? Ethanol content was not determined?
The Results discussion needs to take into account the presence of all microorganisms during fermentation.
Minor observations:
1. lines 77-78-please correct the word temperature
2. line 82 please correct the word character
3. line 96 is Boehringer the correct word?
4. line 70-please indicate the used ratio between defective banana fruits and banana flowers
5. line 130-please indicate producer, city, country for vortex shaker
6. line 153-please indicate producer, city, country for the oven
7. line 270-please correct the word esiling
8. line 346-please correct the word improve
Author Response
Dear Editor and Reviewers
Thank you very much for evaluating our paper.
We have revised the manuscript “Characterization of Lactic Acid Bacteria Isolated from Banana and Application for the Silage Fermentation of Defective Banana” carefully based on the comments of Editor and Reviewers.
In this revised version, changes to our manuscript were all highlighted within the document by using red colored text. We hope that our paper much better quality than before.
The corresponding responses to the reviewers are as followings:
Reviewer 2
Comments and Suggestions for Authors
The manuscript: Characterization of Lactic Acid Bacteria Isolated from Banana and Application for the Silage Fermentation of Defective Banana, present very interesting results from field of microbiology and silage preservation. The use of by-product in animal nutrition is increasing and the data presented in this manuscript will be of great impact in banana producing countries. But also in other regions with high volume of unused by-product in agriculture.
Response: Thank you for evaluating our paper. We will do the best to improve our manuscript.
Comments to authors:
Line 121, delete: then the analysis was done
Response: Yes, we deleted “then the analysis was done” in the text.
Line 293 to 307- part of the text describing factors affecting silage fermentation and factor involved in assessing good silage could be transferred to introduction section and maybe expand bit. This will also improve the introduction section.
Response: Thanks for your comments, here is the discussion for the Results section. We use parts of it to reinforce the introduction section.
Line 350 to 357 are more like conclusion text. The text could be added accordingly to the conclusion section
Response: Yes, we have rearranged and added to the conclusion section accordingly.

Reviewer 3 Report
The manuscript: Characterization of Lactic Acid Bacteria Isolated from Banana and Application for the Silage Fermentation of Defective Banana, present very interesting results from field of microbiology and silage preservation. The use of by-product in animal nutrition is increasing and the data presented in this manuscript will be of great impact in banana producing countries. But also in other regions with high volume of unused by-product in agriculture.
Comments to authors:
Line 121, delete: then the analysis was done
Line 293 to 307- part of the text describing factors affecting silage fermentation and factor involved in assessing good silage could be transferred to introduction section and maybe expand bit. This will also improve the introduction section.
Line 350 to 357 are more like conclusion text. The text could be added accordingly to the conclusion section
Author Response
Dear Editor and Reviewers
Thank you very much for evaluating our paper.
We have revised the manuscript “Characterization of Lactic Acid Bacteria Isolated from Banana and Application for the Silage Fermentation of Defective Banana” carefully based on the comments of Editor and Reviewers.
In this revised version, changes to our manuscript were all highlighted within the document by using red colored text. We hope that our paper much better quality than before.
The corresponding responses to the reviewers are as followings:
Reviewer 1
The article “Characterization of Lactic Acid Bacteria Isolated from Banana and Application for the Silage Fermentation of Defective Banana” is focused on studying the efficiency of lactic fermentation of banana by-products (defective banana) in order to obtain silage.
The topic of the paper is of relevance due to the current increasing trend of valorization of different by-products from fruit production and processing.
The study presents numerous analyses and results but the submitted manuscript revealed some inconsistencies which have to be corrected before publication.
Response: Thank you for evaluating our paper. We will do our best to revise this manuscript.
- The aim of the manuscript needs to be rephrased to better indicate the research purpose.
Response: Yes, we have rewritten the purpose according to your comments in the introduction section.
- It is not clearly explained why the authors used sucrose and tannase in their experiment. I suggest introducing an explanatory paragraph to the Introduction section. Moreover, Lb.plantarum did not metabolize very well sucrose, especially when the substrate contain other simple sugars.
Response: Yes, based on your suggestion, we clearly explained why sucrose and tannase were used in the experiment in the introduction section.
- I suggest introducing to the Materials and Methods section a graphical scheme of the experiment to be easier understood by readers.
Response: Yes, we have added the experimental process to the experimental section, hoping to better understand the experimental method.
- Some results are poorly discussed; for example fiber content
Response: Yes, bananas are rich in nutrients. In addition to CP and fat, they are also rich in plant dietary fiber, minerals and vitamins. The content of these useful components in bananas, the dynamic changes during silage fermentation and the response to the physiological metabolism of livestock are necessary to be further explored in future studies. We have addressed this in the discussion section.
- As is shown in Table 3, during ensiling not only LAB but also yeasts and aerobic bacteria are present. Are these microorganisms involved in fermentation? For example, yeast count is relatively high during the first 10 days; yeast cells
Response: Thank you for your very important question. Yes, these microorganisms are involved in the fermentation process and are harmful microorganisms for silage fermentation. We describe it in more detail in the Discussion section.
Minor observations:
- lines 77-78-please correct the word temperature
Response: Yes, we have revised.
- line 82 please correct the word character
Response: Yes, we have revised.
- line 96 is Boehringer the correct word?
Response: Yes, we have confirmed that it is correct.
- line 70-please indicate the used ratio between defective banana fruits and banana flowers
Response: Defective bananas are immature and the fruit and flower parts have not been separated. The proportion of flowers and fruits cannot be specified accurately.
- line 130-please indicate producer, city, country for vortex shaker
Response: Yes, we have added the information in the test.
- line 153-please indicate producer, city, country for the oven
Yes, we have added the information in the text.
- line 270-please correct the word esiling
Response: Yes, we have revised as ensiling.
- line 346-please correct the word improve
Response: Yes, we have revised as improve.
Reviewer 2
Comments and Suggestions for Authors
The manuscript: Characterization of Lactic Acid Bacteria Isolated from Banana and Application for the Silage Fermentation of Defective Banana, present very interesting results from field of microbiology and silage preservation. The use of by-product in animal nutrition is increasing and the data presented in this manuscript will be of great impact in banana producing countries. But also in other regions with high volume of unused by-product in agriculture.
Response: Thank you for evaluating our paper. We will do the best to improve our manuscript.
Comments to authors:
Line 121, delete: then the analysis was done
Response: Yes, we deleted “then the analysis was done” in the text.
Line 293 to 307- part of the text describing factors affecting silage fermentation and factor involved in assessing good silage could be transferred to introduction section and maybe expand bit. This will also improve the introduction section.
Response: Thanks for your comments, here is the discussion for the Results section. We use parts of it to reinforce the introduction section.
Line 350 to 357 are more like conclusion text. The text could be added accordingly to the conclusion section
Response: Yes, we have rearranged and added to the conclusion section accordingly.
Academic Editor Comments
I agree with the author’s opinion that the genome sequencing of the strain is of little meaning to their work, and the characterization of the lactic acid bacteria based on the comprehensive analysis of its physiological and biochemical properties and 16S are sufficient for the purpose of this study. I, however, did not find the accession numbers of 16S rRNA sequences of lactic acid bacteria in the Materials and Methods section. In “Instructions for Authors” of Microorganisms, it is mentioned that “Accession numbers of RNA, DNA and protein sequences used in the manuscript should be provided in the Materials and Methods section.” Please provide accession numbers of 16S rRNA sequences of your bacterial strains.
Response: Thanks for your comments. We have registered the 16S rRNA sequences of the strains in GenBank and entered the accession numbers in the methods section.
Thank you for the all comments.
Best regards.
Dr. Yimin Cai
Japan International Research Center for Agricultural Science (JIRCAS)
Tsukuba, Ibaraki 305-8686, Japan

Round 2
Reviewer 1 Report
I couldnot find any response to my previous comments as follows:
he work is interesting, valuable, well designed and well written.
Major concern
The most dangerous point in this work is the final approval for strains to be used in silage as D-Lactate has a direct neurotoxic effect and various health problems please kindly refer to https://www.hindawi.com/journals/bmri/2020/3419034/
The authors should deeply discuss this point and confirm the validity of the final end product, otherwise, this Silage product would be toxic.
Minor comment
Please update the data at line 36-37
Author Response
Reviewer 1
Comments and Suggestions for Authors
I couldnot find any response to my previous comments as follows:
he work is interesting, valuable, well designed and well written.
Response: Thank you for evaluating our paper. We are really sorry that the order of reviewers has been wrong. We have restructured and responded as follows:
Major concern
The most dangerous point in this work is the final approval for strains to be used in silage as D-Lactate has a direct neurotoxic effect and various health problems please kindly refer to https://www.hindawi.com/journals/bmri/2020/3419034/
The authors should deeply discuss this point and confirm the validity of the final end product, otherwise, this Silage product would be toxic.
Response: Thank you for the comments and information. This is a very important question, and the authors have studied the effects of lactate isoforms of feed on the physiological metabolism of livestock in previous studies. Based on your suggestion, we discuss the lactic acid isoforms produced by lactic acid bacteria in silage and their influence on livestock in the discussion section, and added related literature.
“Two isomers of lactic acid, L-lactic acid and D-lactic acid are produced by LAB in silage fermentation. In animal, the L-lactic acid is more effectively utilized within the body than the D-lactic acid, while the D-lactic acid has no direct neurotoxic effect on livestock. In the production of livestock feed, L-lactic acid producing Lactobacillus casei and DL-lactic acid producing Lactiplantibacillus plantarum are widely used to develop the commercial inoculant. The LAB screened in this study is Lactiplantibacillus plantarum, forms DL-lactic acid. In the process of silage fermentation, this LAB can produce a large amount of lactic acid and re-duce pH during ensiling, thereby improving the fermentation quality of silage.”
Minor comment
Please update the data at line 36-37
Response: Yes, we updated these data on lines 36-37.

Reviewer 2 Report
I agree with this form of the manuscript
Author Response
Thank you for rating our paper and we upload a new version of the manuscript.
